# Understanding the mental health impact and needs of public healthcare professionals during COVID-19 in Pakistan : a qualitative study

Waqas Hameed [1], Anam Shahil Feroz,[1] Bilal Iqbal Avan [2], Bushra Khan,[3] Zafar Fatmi [1], Noreen Afzal,[1] Hussain Jafri,[4] Mansoor Ali Wassan,[5] Sameen Siddiqi [1]

[1]Department of Community Health Sciences, Aga Khan University, Karachi, Pakistan
[2]Department of Population Health, London School of Hygiene and Tropical Medicine, London, UK
[3]Department of Psychology, University of Karachi, Karachi, Pakistan
[4]Punjab Thalassaemia and other Genetic Disorders Prevention and Research Institute, Fatima Jinnah Medical University, Lahore, Pakistan
[5]Health Department, Government of Sindh, Karachi, Pakistan

**Correspondence to**
Dr Bilal Iqbal Avan;
Bilal.Avan@lshtm.ac.uk

## ABSTRACT

**Objectives** A dearth of qualitative studies constrains in-depth understanding of health service providers' perspectives and experiences regarding the impact of COVID-19 on their mental health. This study explored the mental health impact and needs of of public sector healthcare workers during COVID-19 who working in secondary-level and tertiary-level healthcare settings of Pakistan.

**Design** An exploratory qualitative study.

**Setting** Twenty-five secondary-level and eight tertiary-level public hospitals of Sindh and Punjab provinces of Pakistan.

**Participants** In-depth interviews were conducted with 16 health service providers and 40 administrative personnel. Study data were analysed on NVivo V.11 using the conventional content analysis technique.

**Results** The study identified three overarching themes: (1) mental health impact of COVID-19 on health service providers that included the fear of acquiring the infection and transmitting it to their family members, fear of social isolation and stigma, anxiety related to the uncertainty of COVID-19, nervousness due to media exaggeration and stress associated with excessive workload; (2) mental health needs of health service providers involved in the COVID-19 crisis and available support from the healthcare system, including the expression of the need for counselling services and safe working conditions, the need for paid rest periods, and the need for appreciation and motivation to work in the pandemic; and (3) suggestions to address mental health needs of healthcare workers, including provision of specialised mental healthcare/ services, formal training of health managers on managing mental health needs of health facility staff, and assessment and addressing of these needs of the health workforce.

**Conclusion** The study emphasises the need to strengthen health system preparedness for recognising and addressing the needs of healthcare professionals. At the system level, there is a need for a specialised unit to provide mental health services and better communication strategies. At the staff level, continuous motivation and appreciation should be given to healthcare professionals either through monetary incentives or formal acknowledgement of their performance.

## STRENGTHS AND LIMITATIONS OF THIS STUDY

⇒ This study provides an in-depth exploration of experiences of healthcare professionals regarding the mental health impact of COVID-19, as well as their mental health needs and suggestions to address these needs in a pandemic crisis.

⇒ Unlike previous urban-focused studies, the participants in this study were invited across 22 urban and rural districts in the two most populous provinces of Pakistan.

⇒ One key limitation of the study was that the interviews were conducted by phone owing to restricted mobility during the COVID-19 pandemic, which may have introduced some risk of bias as the interviewers did not have the opportunity to build rapport as in face-to-face interviews.

## INTRODUCTION

Worldwide, health service providers are under insurmountable psychological pressures as they are expected to deal with a number of issues that arise due to the COVID-19 situation.[1] First, due to the increased clinical workload associated with COVID-19 cases, health service providers face increasingly long work hours, often with limited resources and weak infrastructure.[2] Second, they face physical discomfort associated with donning and doffing of personal protective equipment (PPE), or the unavailability of it, which is essential to keep them safe from exposure to the virus.[3] Third, health service providers feel unprepared to deal with a new and unfamiliar task since the evidence-based clinical treatments for COVID-19 continue to evolve.[3] Finally, there is a very valid fear of autoinoculation and the risk of transmitting the virus to family members.[4][5] Unsurprisingly, all these factors take a toll on the mental health of healthcare professionals (HCPs), and result in various psychiatric issues including fear,[6]

anxiety,[1] depression,[1] burnout,[7] trauma[8] and insomnia.[9] Evidence shows that mental health implications for health workers who are involved in the provision of healthcare during epidemics and pandemics are long-lasting.[10 11]

While quantitative studies have significantly outnumbered qualitative investigations[1 3 6 8 12] in terms of assessing the magnitude of mental health problems among HCPs resulting from COVID-19, they are limited in providing an in-depth understanding of HCPs' perspectives and experiences regarding the impact of COVID-19 on their mental health. Moreover, most qualitative investigations explored the general experiences of HCPs during the COVID-19 pandemic,[13 14] whereas very few have specifically focused on their mental health experiences and these are mainly from high-income countries.[12] A qualitative study from England concluded that patients with COVID-19 brought a significant emotional toll, and strained relationships between immediate frontline staff and their families. At the beginning of the pandemic, staff were driven by adrenalin and optimism; but over time this dissipated to be replaced by exhaustion, numbness and dreadful expectation of a 'second wave'.[15] Another qualitative study from Pakistan[16] presented a thorough understanding of how the frontline emergency healthcare workers (HCWs) are dealing with the COVID-19 pandemic, their stress-coping strategies or protective factors, and challenges while dealing with patients with COVID-19. These studies are limited in providing evidence on health systems' preparedness to respond to the mental health needs of HCPs. Earlier evidence during the SARS epidemic showed that system-level interventions were effective in reducing health service providers' symptoms of anxiety and depression.[1–4 11]

By and large, there is not only a dearth of qualitative studies on mental health impact of COVID-19 on HCPs from low/middle-income countries but there are also gaps in existing literature.[13] For example, there is a lack of information about the psychosocial and mental health needs of HCPs and how well the health system has responded to those needs.[12] Furthermore, these qualitative studies had a predominant focus on hospitals situated in urban settlements or metropolitan cities, while their sample did not have adequate representation from rural settings. Also, the HCPs included in many quantitative studies were mainly doctors and nurses; few studies have gathered perspectives of other HCPs—such as health administrators.[12]

To the best of our knowledge, there is a paucity of qualitative evidence to understand the mental health needs of HCPs and the health system response, which is essential to develop context-specific mental health interventions. This study is the first to explore the mental health impact of COVID-19 as well as the mental health needs of health service providers in public healthcare settings of Pakistan from the perspective of health managers and health service providers including doctors and nurses.

Our study aims to answer the following research questions:

► What is the perceived mental health impact of the COVID-19 pandemic on the performance of health service providers in secondary and tertiary care settings in Pakistan?
► What are the health service providers' mental health needs working amid the COVID-19 pandemic, and how well the health system is responding to them?
► What are the recommendations by the health service providers to address their mental health needs?

## METHODS
### Country context
Pakistan has a population of over 225 million inhabitants.[17] The first COVID-19 case was identified in February 2020, and since then a total number of confirmed cases that exceed 1.5 million along with 30 574 deaths have been registered (until 29 August 2022).[18] The country has experienced five waves—each with a different variant including omicron, more recently.[18] With a focus on vulnerable groups of HCPs and elderly (65+ years) people, the country embarked on a nationwide vaccine campaign for COVID-19.[18]

Healthcare in Pakistan is delivered through a three-tiered health system. Secondary-level healthcare facilities concern with provision of technical, therapeutic and diagnostic services. Specialist consultation and hospital admissions fall into this category. The availability of specialised services vary from facility to facility across districts depending on the catchment population and their service needs and demands.[19] It includes Tehsil Headquarters (THQs) and District Headquarters (DHQs) Hospitals. Tertiary healthcare hospitals are for more specialised inpatient care supported by availability of required resources. With few exceptions, these are also affiliated with the medical teaching institutions for graduates and postgraduates.[20] The government of Pakistan has taken several measures to combat this deadly pneumonia virus[21] but it leaves a lot to be desired.[22] The health system of Pakistan is weak,[23] with critical shortage of health workforce which is not prepared to sustain a major surge in COVID-19 cases.[24]

Recent quantitative studies show the considerable impact of COVID-19 on the psychological health of HCPs.[25 26] Generally in Pakistan, mental illnesses are stigmatised and widely perceived to have supernatural causes. The psychiatrist-to-population ratio is about 1 psychiatrist to 0.5–1 million people[27]; this—combined with limited task-sharing interventions—results in a huge treatment gap in the country. The economic burden of mental illnesses in Pakistan is US$4.2 billion annually.[28]

### Design, setting and participants
The Consolidated criteria for Reporting Qualitative research checklist has been used for designing this qualitative study[29] (see online supplemental file 1).

This research employed an exploratory qualitative research design using semistructured interviews and a criterion purposive sampling approach. Our study was

conducted at secondary-level and tertiary-level public sector hospitals of the two most populous (Sindh and Punjab) provinces of Pakistan. A total of 8 teaching hospitals, 8 DHQs, and 25 THQs were randomly selected from Sindh and Punjab provinces. The study was conducted in close collaboration with provincial health departments.

The data collection methods for this research included in-depth interviews (IDIs) with health service providers (doctors, nurses), head of department (HoD) and medical superintendent (MS). In terms of participants' involvement in the care of patients with COVID-19, health managers including MS and HoD were indirectly involved in the management of patients with COVID-19; however, health service providers including doctors and nurses were directly involved in the care of patients with COVID-19. For secondary-level health facilities, we contacted the MS who was invited to participate in the study. Moreover, he/she was further asked to nominate service providers from their respective health facilities who may or may be not involved in the provision of care to patients with COVID-19. Similarly, for teaching hospitals, we contacted the administrative head who identified a focal person on his/her behalf for further coordination. Similar request was made to the focal person for identification of department heads and service providers. Only nine participants refused to participate mainly due to their busy schedule and lack of interest.

### Eligibility criteria

Health managers and health service providers employed in public sector hospitals of Sindh and Punjab provinces and were dealing directly or indirectly with patients with COVID-19 were eligible to participate in the study. Health service providers and hospital managers either belonging to private sector hospitals or from provinces other than Sindh and Punjab were excluded. Non-clinical staff such as janitors and technicians were also excluded from the study.

### Data collection

Adapting to the COVID-19 situation, a total of 56 (Sindh: 34; Punjab: 22) semistructured telephonic IDIs were conducted individually in the local language (Urdu), by trained female data collectors who had a background in sociology, anthropology and psychology. The interviews are conducted at the Aga Khan University Hospital under close supervision of research investigators. Keynotes were also taken by data collectors during the interviews. It was ensured that the participant is comfortable during the interview and no one is sitting near to them whose presence could possibly influence his/her responses. To ensure data quality, a few initial interviews were conducted by WH (PhD scholar–health system), ASF (PhD scholar–public health) and NA (BS psychology) in the presence of trained data collectors. The IDIs were conducted until data saturation was achieved.[30] Data collection and data analysis were conducted iteratively to determine the point of data saturation.[31 32] All interviews were completed in

the first interaction. Consistent with Creswell and Poth's[31] guidelines on interviewing, all IDIs lasted approximately between 45 and 60 min. The interviews were audio-recorded on tablets with consent from study participants, which were later transcribed in the English language by a transcriptionist. The data collection took place between August and October 2020. The transcripts were not returned to the participant due to their busy schedule in the hospitals.

A semistructured interview guide was designed for IDIs through an extensive literature review and consultative approach seeking expert opinions. This guide was developed to interview all research participants. The guide includes five major themes: (a) mental health impact of COVID-19 on health service providers; (b) mental health needs of health service providers involved in the COVID-19 crisis; (c) support mechanisms available to address mental health needs of health service providers; (d) challenges in addressing mental health needs of health service providers; (e) recommendations to address mental health needs of health service providers. A free flow of information was encouraged, using the participants' verbatim account of the conversation to understand their perception of mental health needs during the COVID-19 pandemic.

The interview guide was used flexibly to allow participants to construct their accounts on their own terms. The guide was pilot tested with a non-study sample which shares the same characteristics as the study sample.[31] The pilot testing provided evidence-based direction to improve the interview guide questions[32] (see online supplemental file 2).

### Data analysis

All study data including notes, recordings and transcriptions were uploaded into the NVivo V.11.0 software for data analysis.[33] The analysis looked at convergent and divergent lines of inquiry from the two main study groups, that is, health managers and health service providers (doctors and nurses). All qualitative data included were analysed using conventional content analysis.[34 35] Two independent researchers (ASF and NA) read each transcript from beginning to end, and coded text that appeared to describe the mental health impact of COVID-19 as well as the mental health needs of health service providers in public healthcare settings of Pakistan. After open coding of initial few transcripts, independent researchers met to discuss and decide on the preliminary codes (codebook). Researchers then coded the remaining transcripts (and recoded the original ones) using the codebook and added new codes when they encountered data that did not fit into an existing code. Once all transcripts had been coded, researchers examined all data within a particular code. Some codes were combined during this process, whereas others were split into subcategories. The discrepancies in coding were resolved through

a discussion with a senior researcher. Throughout the analysis, the researcher made reflective notes (memos) to document important learning from the data.[36] ASF and NA independently performed coding to ensure that interpretations are coherent, plausible and grounded in the study data.[32] Please see online supplemental file 3 to see an example of content analysis.

## Trustworthiness of the study

Of the five markers suggested by Tracy[37] and Lincoln et al[38] to ensure trustworthiness and methodological rigour, we used validity and credibility and reliability/dependability. To ensure credibility, the study triangulated data via two basic types of triangulation: data source triangulation (exploring insights of different groups—hospital managers and health service providers) and investigator triangulation (use of multiple researchers in analysis phase—ASF and NA).[31] In addition, the primary researcher maintained a reflexive journal during all stages of the research to recognise and acknowledge biases during the research process. For example, reflexivity was used to develop the interview guides and modify it based on the inputs from other investigators and after the initial interviews, and analyse the data in different groups to tell a story about how the health system responded to deal with the mental health needs of health service professionals during the COVID-19 pandemic.

## Patient and public involvement

We sought inputs from target audience through pretesting of interview guides to ensure comprehension of the questions. Furthermore, in consultation with provincial health departments, we selected districts where COVID-19 cases were high. The focal person in each selected district was oriented about the study who connected our team with those in charge in the health facility to avoid any miscommunication. Lastly, they were also advised on the dissemination of research findings—that is, they specifically emphasised sharing it through research briefs to reach a wider audience in the country.

## RESULTS

A total of 56 semistructured IDIs were conducted to explore the mental health needs of health service providers amidst the COVID-19 pandemic in Pakistan. Of 56 IDIs, 16 IDIs were conducted with health service providers, and 40 IDIs were conducted with health managers. All the participants (n=56) who were approached by the study team agreed to participate. The demographic information for all the study participants is illustrated in table 1.

Based on the directed content analysis, three overarching themes were identified: (1) mental health impact of COVID-19 on health service providers;

**Table 1** Characteristics of study participants

| Characteristics of study participants | n (%) |
|---|---|
| Gender (n=56) | |
| Female | 9 (16.1) |
| Male | 47 (83.9) |
| Designation (n=56) | |
| Hospital managers | 40 (71.4) |
| Health service providers | 16 (28.5) |
| Facility type (n=56) | |
| Secondary healthcare hospitals | 37 (66.1) |
| Tertiary healthcare hospitals | 19 (33.9) |
| Age (n=48) | |
| Mean | 46.6 |
| Median | 47 |
| SD | 10.6 |
| Experience (n=53) | |
| Mean | 15.4 |
| Median | 15 |
| SD | 9.5 |

(2) mental health needs of health service providers involved in the COVID-19 crisis and available support from the health system; and (3) suggestions to address mental health needs of health service providers. These themes and their categories are presented in table 2.

## Mental health impact of COVID-19 on health service providers
### Fear of infection, isolation and stigma

The interviews with hospital managers and health service providers revealed that all health service providers felt fearful of acquiring the COVID-19 infection and transmitting the infection to their family members. A hospital manager mentioned that health service providers are particularly concerned about their parents and children who are vulnerable and might get infected.

> They (HCPs) were afraid for their families that our parents are old and if we carry coronavirus from here then we might not get them infected. Even if we don't show any symptoms, we would get them infected or we would get our children infected. So this fear was also prevailing. (Hospital Manager, Punjab)

Health service providers were fearful that if they did acquire the infection, they would have to isolate themselves from their families. Many workers could not afford to isolate themselves in the event they tested positive for COVID-19.

> They used to say that if the report comes back positive, then how they will stay at home because their homes are small. (Hospital Manager, Sindh)

> Immediately after that, there was fear of isolation, like if you put yourself into isolation who will look

**Table 2** Themes and subthemes

| Themes | Subthemes |
|---|---|
| Theme 1: mental health impact of COVID-19 on healthcare professionals | 1.1 Fear of infection, isolation and stigma<br>1.2 Stress due to poor availability of personal protective equipment<br>1.3 Anxiety due to uncertainty of COVID-19<br>1.4 Anxiety and nervousness due to media exaggeration<br>1.5 Excessive workload |
| Theme 2: mental health needs of the health workforce involved in the COVID-19 crisis and available support from the health system | 2.1 Need to provide counselling services and safe working conditions<br>2.2 Need for paid rest periods, timely payment of workers' salaries and incentives<br>2.3 Need of appreciation and motivation to work in the pandemic |
| Theme 3: suggestions to address the mental health needs of healthcare professionals | 3.1 Establish mental health services to address the mental needs of health workers<br>3.2 Conduct formal training of health managers in managing mental health needs<br>3.3 Assess and address mental needs of the health workforce |

after your family and your kids. (Hospital Manager, Punjab)

Both groups of respondents highlighted that few health service providers resigned from their jobs since their families were getting infected because of them. In particular, female health service providers had higher pressure on them from their families to leave their jobs. While some providers refused to work in COVID-19 wards, others threatened health managers to commit suicide, if they were posted there. Health managers verbalised that a lot of the health service providers posted in COVID-19 wards revealed the fear of being isolated and stigmatised.

Health workers were hospital-bound and were afraid of living alone and being cut off from their families. They also felt the stigma from the people around them. (Hospital Manager, Sindh)

### Stress due to poor availability of PPE

A major concern among the health service providers was the lack of protective gear available to them. Thus, treating COVID-19-positive or suspected cases without PPE was escalating their fear of getting infected. They felt that they had been sent to fight a war without weapons.

Firstly, the PPEs we were using, we had to use them twice or thrice… this was our main issue. We had to wear the mask for two to three days even one week continuously. (Health Service Provider, Punjab)

On the other hand, health managers revealed that some health service providers labelled them as being insensitive and unaware of the problems faced by workers due to a lack of available PPE. However, managers themselves were helpless due to PPE's lack of availability in the market and ended up buying masks and other protective equipment for the staff themselves, but this protective gear was not up to the mark.

That time if anyone didn't get PPE they (health service providers) would get hyper but we provided PPEs

at our facility. We would provide requisite equipment soon so we didn't have to face so much trouble. (Hospital Manager, Punjab)

### Anxiety due to uncertainty of COVID-19

Health service providers reported feeling extremely anxious working with patients with COVID-19. Owing to the novelty and unfamiliarity of the virus, along with workers being inadequately equipped with the required knowledge about the disease, they were uncertain about the consequences. Health service providers were also having trouble identifying cases of COVID-19 since there were both symptomatic and asymptomatic cases. They were not clear about the mode of transmission of the virus.

One can call it anxiety, you can see it, and they were under a depressive state. All healthcare workers were. From doctors to nursing staff, paramedics, our sweepers, all our workers were going through depression. (Health Service Provider, Sindh)

Some hospital managers had contrasting views about the difficulties faced by their staff while working during the pandemic. They claimed that health service providers were not stressed while working during the COVID-19 pandemic and that none of them came to the managers to express their concerns.

I don't think their mental health was affected, I don't think so, and there was stress only in the beginning when they were not aware of the atmosphere. (Hospital Manager, Sindh)

No there was no pressure on them, we were doing many meetings, we were explaining it to them, that there's no issue, this time has come we will solve it Insha'Allah, we will be successful, there wasn't an issue like this in our staff, they were definitely cooperating with us, that's why there was no stress or any problem like that. (Hospital Manager, Sindh)

### Anxiety and nervousness due to media exaggeration

Both groups of respondents reported that the media has a strong role in mounting anxiety and distress among HCWs. The media was seen to exaggerate the magnitude of the situation and there was a bombardment of information. News regarding deaths of workers due to COVID-19 was also creating panic among the majority of the HCWs.

> But this novel coronavirus made hype through digital media as no one knew what it was as initially people were stressed. (Hospital Manager, Sindh)

False information was seen to circulate about the negative consequences for people testing positive for COVID-19, which was creating a lot of distress among HCWs currently working in COVID-19 wards. In addition, news related to the long-term complications of COVID-19 increased nervousness among HCWs.

> If you take my personal opinion, the hype created regarding this is too much, that has spread too much tension. That has bothered people a lot about what will happen, God forbid if somebody has corona then he will be taken away or this will happen or that will happen, etc or because of me, my children will suffer. (Health Service Provider, Punjab)

### Excessive workload

Both groups of respondents mentioned that health service providers felt strained and exhausted because of the clinical workload. Due to the shortage of health workforce, the existing pool of health service providers had to bear the burden of extra clinical work. Their duty hours increased along with the number of shifts they had to do per week, which resulted in increased psychological distress among health service providers.

> So far, no one told me that their mental health has been affected… yes, some people have said that they are depressed… they are feeling tired and have motion sickness due to extra working hours. (Health Manager, Punjab)

The COVID-19 outbreak among health service providers led to further staff shortages and increased mental pressure among health service providers. Both groups of respondents mentioned that even after returning from sick leave, health service providers were not performing at their best. Some of them seem to become weak and some of them would faint. In addition, some workers' tests were negative for the COVID-19 infection, but they still had cough symptoms which would make their coworkers anxious. Hence, they also had to be given leave.

> As the work load increased, so did the pressure on the HCWs and their efficiency began to decrease. (Health Manager, Sindh)

Both groups of respondents mentioned that their commute time to the hospital also significantly increased due to the unavailability of public transport during the lockdown, which further increased the psychological burden among health service providers.

> Our doctors faced a lot of transport issues but we tried our best to support them. Few of our staff were punctual initially, but some of them were not coming at all due to fear of COVID-19. Their wages are less so we reduced the duration of their duties and counselled them a lot after which they started coming to work. They come from far off areas so they should be given space here at the hospital so that they can perform their duties easily. (Health Manager, Sindh)

### Psychological needs of the health workforce involved in the COVID-19 crisis and available support from the health system
#### Need to provide counselling services and safe working conditions

Health service providers and health managers highlighted the need for counselling services for health service providers working under pressure to provide care to patients with COVID-19. Health service providers mentioned the desire to receive one-to-one counselling sessions from mental health professionals to reduce the stress and anxiety they were experiencing because of working during the COVID-19 pandemic.

> All hospitals should at least provide one-to-one… counselling sessions… to reduce staff anxiety and stress. (Health Service Provider, Sindh, male)

Since no proper counselling services from mental health professionals were available, informal counselling to health service providers was provided by hospital managers. Although the managers were trying to support their staff, they lacked formal training in providing counselling. They motivated them via reinforcing health providers' professional and moral obligations but it did not prove fruitful for most of the workers.

> Whatever knowledge I had regarding the disease and its spread, and I also told them that we must serve humanity, and death is inevitable and I did whatever I could, regarding the moral values, whatever I could make them understand, some of them understood and most of them didn't. (Hospital Manager, Sindh)

> I told them with a passion that the death Allah has written, if we die on this job, they will be martyred. (Hospital Manager, Punjab)

Health managers from few facilities reported that the government had arranged webinars for health service providers to help them manage psychological stress. However, the effectiveness of this training was quite low because many health workers had duties due to which they were unable to attend or sit through training.

In addition, health service providers highlighted the need of working in a safe and secure working environment. They further mentioned that the lack of safety was largely attributed to the inadequate supply of protective equipment. To make up for the lack of PPE, health service

providers resorted to using the same protective gear they had repeatedly used for multiple days.

> In many places, we have seen that doctors and paramedical staff were affected and all this is due to the lack of what was needed, for example, the lack of hand sanitizers and masks. (Hospital Manager, Sindh)

### Need for paid rest periods, timely payment of workers' salaries and incentives

Health service providers also highlighted the need for extra rest periods so that they could get enough time to relax from the exhaustive routine and then come back to work. A common strategy employed by the health managers was to give short leave to the health service providers most affected by the COVID-19 stress. This leave helped the staff in regaining their mental and physical strength to work.

> They (health service providers) used to say we will do work for 15 days and then require 15 days for holidays, we will relax during that time, I said ok I will give that to you. (Hospital Manager, Sindh)

In addition, the majority of health service providers and health managers reported that the lack of additional allowance for staff directly dealing with patients with COVID-19 was a demotivating factor especially for paramedical staff.

> The poor paramedical staff who are the first ones to interact with the patients get paid very less. The government did not do anything for them because the staff used to complain that we are getting the same salaries while having maximum exposure to COVID-19. (Health Service Provider, Punjab)

Health service providers also wanted to be given monetary incentives because working during the pandemic meant that their lives were at stake. This would also help to increase their motivation to work during such difficult conditions.

> They should be given incentive because they are keeping their life on risk. (Health Service Provider, Sindh)

### Need for appreciation and motivation to work in the pandemic

Hospital managers and health service providers reported that workers needed appreciation for all the hard work they were doing. Moreover, providing appreciation would, in turn, lead to the workers becoming motivated to continue their efforts. Health service providers were putting their lives at stake while working in such difficult conditions without adequate resources, hence their efforts should be commended.

> Then they (health managers) should arrange programs in which they appreciate and motivate their employees and provide them proper guidance then

it will have major positive effects on mental health. (Health Service Provider, Punjab)

It was also suggested that the staff should be given certificates to appreciate the work they were doing and to hold regular meetings with them to see work progress and understand their needs. Health service providers mentioned that motivating sessions are also needed to boost morale of the workers who are working in such difficult conditions.

> I told my hospital director that we need a psychologist. I also highlighted that our human resource team should be given appreciation certificates other than monetary incentives which might boost their motivation to work. Meetings should also be conducted with them to ask them about their work and what else they need since this is also lacking currently. (Hospital Manager, Sindh)

### Suggestions to address mental health needs of HCPs
### Establish mental health services to address the mental health needs of health workers

Both groups of respondents stressed that there needed to be an increase in the budget allocated to the health sector to establish mental health services for health service providers working during the COVID-19 pandemic. More funds and resources would lead to the betterment of HCWs. Hiring mental health professionals including psychologists and psychiatrists to support health service providers psychologically was strongly recommended and emphasised repeatedly by health managers.

With regard to the roles of provincial governments in improving the mental health of HCPs, there was a need for the implementation of formal programmes addressing mental health. The government was also advised to arrange seminars on mental health for health service providers alongside launching a mobile application for providing relevant information on the topic.

> There should be seminars and sessions and the Punjab government should launch an app regarding this so that we get information from it regarding mental and physical health and keep on updating it. This will improve the situation. (Health Service Provider, Punjab)

Hospital managers were also of the view that government officials should visit hospitals and have discussions with all levels of health service providers to understand their experiences and needs. This would enable the government to formulate effective strategies in dealing with problems faced by health service providers. Some health service providers complained that their health managers were not bothered if their staff were facing any issues.

> Nobody bothers to discuss any issue faced by workers. They only give orders to perform their duties. Our hospital managers, they don't have any concerns

if workers are having any problems or not. (Health Service Provider, Sindh)

Sorting logistical issues such as providing transportation facilities for health service providers to and from quarantine centres was also stressed upon.

### Conduct formal training of health managers in managing mental health needs

Since hospital managers were directly overseeing the health service providers and had the most interaction with them, their formal training in providing basic counselling to the staff was deemed necessary. Health service providers wanted managers to hold daily meetings with them where they could discuss the difficulties they were facing while caring for COVID-19-positive patients.

The administrator of the hospital should talk about it from every doctor. He should do meeting with them to see if they face any mental pressure or not and if he has so the problem then he should sort out his problem. (Health Service Provider, Sindh)

Some health service providers also remarked that hospital managers should be more approachable so that the staff feel comfortable in sharing their concerns with them.

If employees are guided properly in monthly or weekly meetings, their issues are being heard whatever they are. There should be less difference of job between an employee and MS or head of the department so that he can go and discuss his problems with him, so many of problems can be solved. (Health Service Provider, Punjab)

### Assess and address mental health needs of the health workforce

Another recommendation for the future was for the government and health managers to plan all the measures the health system would need to take in the event of such a crisis. They should have observed the situation in countries such as China where the outbreak first began and started thinking of strategies to get a better grip of the impending problems.

We don't know if this pandemic will subside or not. God Forbid if this second waves comes and traffic of patients increases, then what will we do? This way we will get depressed, so proper training should definitely be given, mental health workers should come, go to the institution and guide the HCWs. (Health Manager, Sindh)

Health service providers also suggested that keeping in mind the difficult conditions they were eventually going to have to work in, there should have been sessions that would mentally prepare them for these difficult conditions.

When you send a doctor in the COVID-19 ward, there should have been a week-long training with a psychologist, to explain them the situation, what will be their duties, the government has selected you, for this task, they should have been prepared and there is nothing to worry about, the mortality rate is low, and most people recover. (Health Service Provider, Sindh)

## DISCUSSION

The large-scale unprecedented public health catastrophe, in the form of COVID-19, placed the frontline HCPs, who are directly involved with all aspects of care of patients with COVID-19, at risk of developing psychological distress and other mental health symptoms.[39] Our study provided an in-depth investigation into the mental health impact of COVID-19 on HCPs and mental health needs of different cadres (health service providers and administrators) in public health sector settings.

We documented an adverse perceived mental healht impact experienced by health service providers while working during the pandemic as revealed in other studies from other countries.[12 40] The health service providers expressed several explanations for psychological stress including the fear of acquiring the infection and transmitting it to their family members, fear of social isolation and stigma, anxiety related to the uncertainty of COVID-19, nervousness due to media exaggeration and stress associated with excessive workload. Evidence shows that exaggeration by the media tends to negatively affect the mental health of HCPs.[41 42] Similarly, lack of knowledge about COVID-19 infection is likely to increase anxiety/depression among physicians[43] and inadequate precautionary measures in the workplace cause fear and anxiety among HCPs.[44] Furthermore, overburdening due to high intensity of care and increased volume of clinical services has been associated with work-related stress, burnout and post-traumatic stress disorder, which in turn negatively affect job performance and quality of services.[42 45]

Perceptions of the two groups of respondents (service providers and health managers) on most issues were largely similar. However, at some instances, we noted divergent views among the two groups regarding the impact of COVID-19 on the mental health of health service providers. It is worrisome to note that while health service providers claimed to experience significant adverse psychological effects due to COVID-19, some health managers denied that their staff faced any stress or difficulty working during the pandemic. These contradictory views may be attributed to a disconnect or lack of communication between service providers and health managers. Another reason for this denial from health managers could be an attempt to express that things are managed well by them since they are generally responsible for the safety and well-being of service providers. Nonetheless, it is important to investigate the causes of such contrasting views about the same phenomenon by

 Hameed W, *et al. BMJ Open* 2022;**12**:e061482. doi:10.1136/bmjopen-2022-061482

different groups, especially when it affects the health system at large.

The study revealed the mental health needs of health service providers including the need to provide counselling services and safe working conditions, the need for paid rest periods and timely payment of workers' salaries, and the need for appreciation and motivation to work in the pandemic. This finding is consistent with other studies where HCWs expressed similar needs.[46 47] For example, previous experience from the Ebola outbreak in western Africa highlighted the need to provide risk allowance for motivating and retaining HCWs.[7] Our study also highlighted the need to provide risk allowance to health service providers. In addition, our study emphasised the need to provide timely salaries to health service providers and extra rest periods. Both groups of respondents revealed that the health system response in the given scenario has been weak given the lack of existing support mechanisms to address the mental health needs of the health workforce during the COVID-19 pandemic. Inefficient management of the pandemic from the healthcare system was also reported in other countries.[40 48 49]

Finally, the health service providers and health managers provided three suggestions to address the mental health needs of health service providers in the future including the provision of mental health services to address the mental needs of health service providers, and conducting formal training of health managers in managing and anticipating the mental health needs of the health workforce to guide future initiatives. Experiences from similar outbreaks suggest that an early mental health intervention and the establishment of early support systems are particularly important to promote the emotional release and improve health service providers' well-being.[50] Similar interventions have been used in the past to address mental health issues in HCWs during an infection disease outbreak.[51] Capacity building of health facility staff on the provision of psychosocial support is also recommended in the WHO (2020) interim guidance document on COVID-19.[42] Call centres have been established in many countries—including Pakistan[52]—to provide needs-based mental health support to HCPs. In Pakistan, the 'WeCare' Programme probably needs to expand its scale of services to effectively address the psychosocial and mental health needs of HCPs. Our study will use knowledge translation tools to share the key findings of this qualitative study with a diverse audience including policymakers, knowledge users, clinicians, researchers, health service providers and hospital managers. Concerning future research direction, insights from the qualitative study would help in streamlining health system response in addressing the mental needs of the health workforce working during the COVID-19 pandemic.

## Strengths and limitations

The study has several strengths and limitations. First, the study used multiple cadres of the health system (health service providers and health managers), which allowed researchers to look for converging and diverging lines of inquiry to identify common themes/concepts and incongruence between data sources. Second, our study sample provides adequate representation from rural settings in the country. Third, the study was conducted in close collaboration with the provincial governments. However, this study is also subject to limitations. First, the study was conducted with a focus on public sector hospitals in only two provinces of Pakistan (Sindh and Punjab), which would limit the transferability of the findings to other provinces.[53] Guba and Lincoln argue for the concept of 'fittingness', which emphasises analysing the degree to which a situation studied matches other situations in which one is interested, and provides a more workable way of thinking about transferability.[53] Thus, this study may provide insights for similar public hospitals across Pakistan that are interested in understanding the mental health impact and mental health needs of health service providers during the COVID-19 pandemic. Second, the study was unable to carry out member checking with study participants, which may have affected the validity of study findings. Lastly, the authors did not have the opportunity to build rapport and obtain non-verbal cues during interviews since the participants were interviewed telephonically due to the COVID-19 situation. In addition, the interviewees might have been distracted during interviews by the activities in their environment and might have experienced fatigue due to long telephonic interviews.

## CONCLUSION

Our findings show that the health system of Pakistan was not prepared to deal with the mental health needs of health service professionals during the COVID-19 pandemic. This information is crucial in anticipating and preparing for tackling possible future pandemics. Health managers and health service providers shared similar concerns about working during the outbreak. Multiple recommendations are made to the government stakeholders for initiating nationwide initiatives for catering to the mental health needs of HCPs. At the system, (a) communication strategies should be devised to ensure better communication between health managers and health service providers at all hierarchies so that the needs and concerns are timely heard and taken care of; and (b) there is a need to establish a specialised unit that should provide mental health services to mitigate the detrimental effects of COVID-19 on the mental health of HCPs. At the staff level, HCPs should be given recognition to boost their morale either through monetary incentives or through formal appreciation based on their performance.

**Acknowledgements** We are grateful to our implementing partners: Department of Health, Government of Sindh; Primary and Secondary Healthcare Department (P&SHD) and Specialized Healthcare & Medical Education Department from the Government of Punjab for extending their full support and guidance to conduct this survey. We are also thankful to all our study participants for sparing their precious

time and made themselves available for the interviews. Last but not the least, we thank our data collectors and project management team: Zafar Dehraj, Imran Sheikh, Sajid Brohi and Ghani Muhammad who managed the implementation of this study in a proficient manner during the pandemic.

**Contributors** WH and BA conceptualised the study with inputs from BK, ZF and SS. WH, BA, BK, ZF and ASF developed the interview guide. HJ and MAW facilitated the study conduct. ASF and NA performed data analysis, supported by WH. WH and ASF took the lead in writing the manuscript. All authors critically reviewed the manuscript and provided feedback in shaping the write-up, analysis and presentation of results. WH addressed comments from coauthors and finalised the manuscript. WH is the overall content guarantor.

**Funding** This work was co-funded by the Aga Khan University, Pakistan (ID: 20051) and the WHO (ID: 202568710-1).

**Disclaimer** The views expressed in this manuscript are those of the authors and contributors, and do not necessarily reflect those of the WHO or the organisation to which the authors are affiliated.

**Competing interests** None declared.

**Patient and public involvement** Patients and/or the public were involved in the design, or conduct, or reporting, or dissemination plans of this research. Refer to the Methods section for further details.

**Patient consent for publication** Obtained.

**Ethics approval** Ethical approval for this study was obtained from the Aga Khan University Ethical Review Committee (AKU-ERC; 2020-5186-11602) and National Bioethics Committee of Pakistan (4-87/COVID-45/NBC/20/393).

**Provenance and peer review** Not commissioned; externally peer reviewed.

**Data availability statement** No data are available.

**ORCID iDs**
Waqas Hameed http://orcid.org/0000-0002-8100-9474
Bilal Iqbal Avan http://orcid.org/0000-0003-4531-4508
Zafar Fatmi http://orcid.org/0000-0001-7212-6858
Sameen Siddiqi http://orcid.org/0000-0001-8289-0964

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
