## [Reviewer comments · BMJ Open]

ARTICLE DETAILS

TITLE (PROVISIONAL)	Understanding the psychological impacts of COVID-19 on public-sector healthcare professionals in Pakistan and exploring their psychological support needs from the health system: a qualitative study
AUTHORS	Hameed, Waqas; Feroz, Anam Shahil; Avan, B; Khan, Bushra; Fatmi, Zafar; Afzal, Noreen; Jafri, Hussain; Wassan, Mansoor Ali; Siddiqi, Sameen

VERSION 1 – REVIEW

REVIEWER	Mostafa Bijani Fasa University of Medical Science, Department of Medical Surgical Nursing, Fasa University of Medical Sciences, Fasa, Iran
REVIEW RETURNED	11-May-2022

GENERAL COMMENTS	Dear Editor, Thank you very much again for the opportunity to review this paper. Introduction, it does not have good coherence. The necessity of the study has not been specified. Why are qualitative studies used? Method: Which conventional content analysis method is used? More details need to be provided on how to encode and extract themes. Please provide an example of theme coding and extraction in a table. Please provide an example of theme coding and extraction in a table. How is the Rigor of qualitative data determined? The discussions have not based on the findings in this study. The discussion section need to be revised to better fit with the findings. It is recommended to use the results of the following studies to develop and strengthen the discussion: 1. Caregivers' perception of the caring challenges in coronavirus crisis (COVID-19): a qualitative study2. Healthcare professionals' families' perception of life during3. COVID-19 in Iran: a qualitative study4. Exploring senior managers' perceptions of the COVID-19 Crisis in Iran: a qualitative content analysis study5. Exploring the experiences of operating room health care professionals' from the challenges of the COVID-19 pandemic6. Health professionals' perception of psychological safety in patients with coronavirus (Covid-19)
--

REVIEWER	Michelle Zechner Rutgers University
REVIEW RETURNED	01-Jul-2022

GENERAL COMMENTS	Overall, this is an interesting paper in a unique population of healthcare providers and an important topic. I think the authors could
--

	further emphasize the importance of gathering perspectives from Pakistani healthcare providers. The paper would benefit from additional editorial review for grammar. Some comments - Sampling - many more administrators than direct care workers - was there a reason for this? Also, the demographics of participants should be noted as a potential limitation - mostly male, hospital managers and secondary healthcare hospitals. Study design - a more comprehensive description of the secondary healthcare systems would be beneficial to the reader. For example, what types of specialists? And are hospital admissions the same as a hospital? This lacks clarity for readers unfamiliar with the healthcare system. p.5 line 50 “secondary and TERRITORY-level” should this be “tertiary” instead? Interviews - were the interviews conducted individually? please clarify why the provinces of Sindh and Punjab were targeted? - How was accuracy of translation into English tested? Quotes - consider placing in a table to reduce length -unclear why some quotes are italicized and others are not -consider inserting brackets [] in the quotes with words to aid the reader understanding (e.g. they would get hyper - is this upset or nervous?) It may be helpful to the reader to have a better understanding of the mental health of healthcare workers in Pakistan. What are the attitudes about mental health support among health professionals? p. 6 line 16 should “Punjab providence” read “Punjab province”? Discussion - It may be helpful to emphasize the importance of workforce attitudes and expectations within Pakistan and how to deliver culturally acceptable workforce supports.
--	---

VERSION 1 – AUTHOR RESPONSE

Reviewer 1 Comments to the Authors:

Dr. Mostafa Bijani, Fasa University of Medical Science

Comments to the Author:

Dear Editor, Thank you very much again for the opportunity to review this paper.

Comment 2.1: Introduction, it does not have good coherence. The necessity of the study has not been specified. Why are qualitative studies used?

Response 2.1: Thank you for your comment. We have revised the introduction section to make it more cohesive and to clearly outline the rationale for our study. A few key reasons to rationalise our study are as follows: a) there is dearth of qualitative study on psychological impact of COVID19 on health professionals from LMICs, b) in addition to psychological impact of COVID19 on HCPs, very few studies have explored the psychological needs of health professional and how well the health system has responded to those needs, c) most studies focuses on hospitals that located in urban settings or metropolitan cities, while not having adequate representation from rural settings, d) based on scoping review, few studies have gather experiences of HCWs other than doctors and nurses (e.g. health administrators), and finally e) there is no qualitative study previously conducted in Pakistan that is similar to our study. Please see revision on 4 and 5.

Comment 2.2a: Methods: Which conventional content analysis method is used? More details need to be provided on how to encode and extract themes. Please provide an example of theme coding and extraction in a table. Please provide an example of theme coding and extraction in a table. Added in the appendix

Response 2.2a: We have elaborated data analysis section. As suggested a sample table has been appended as a supplementary file. Please see page 8 of the manuscript.

Comment 2.2b: How is the Rigor of qualitative data determined?

Response 2.2b: We have added a separate section on “trustworthiness of the study” on page 8 and 9 of the manuscript.

Comment 2.3: The discussions have not based on the findings in this study. The discussion section need to be revised to better fit with the findings.

It is recommended to use the results of the following studies to develop and strengthen the discussion:

1. Caregivers' perception of the caring challenges in coronavirus crisis (COVID-19): a qualitative study
2. Healthcare professionals' families' perception of life during
3. COVID-19 in Iran: a qualitative study
4. Exploring senior managers' perceptions of the COVID-19 Crisis in Iran: a qualitative content analysis study
5. Exploring the experiences of operating room health care professionals' from the challenges of the COVID-19 pandemic
6. Health professionals' perception of psychological safety in patients with coronavirus (Covid-19)

Response 2.3: Thank you for your comment and recommending relevant papers to strengthen the write-up of discussion section. We have thoroughly revised the discussion and organised it according the main themes. Please see page 15-17.

Secondary level minimum health services delivery package for secondary care hospitals (MHSDP)

Reviewer 2 Comments to the Authors:

Dr. Michelle Zechner, Rutgers University

Comments to the Author:

Comment 3.1: Overall, this is an interesting paper in a unique population of healthcare providers and an important topic. I think the authors could further emphasize the importance of gathering perspectives from Pakistani healthcare providers.

Response 3.1: Thank you. We have added country context and shaped the discussion section a bit in accordance with the Pakistani context.

Comment 3.2: The paper would benefit from additional editorial review for grammar.

Response 3.2: Thank you. The manuscript has been proofread by a professional native English.

Comment 3.3: Some comments - Sampling - many more administrators than direct care workers - was there a reason for this? Also, the demographics of participants should be noted as a potential limitation - mostly male, hospital managers and secondary healthcare hospitals.

Response 3.3: Thank you. The reason for higher number of administrators is due to the fact that the larger study was a mixed-method study in which health service providers were mainly included in the quantitative assessment of mental health problems. Therefore, we decided to have fewer of them in qualitative and gathered opinions of administrators. Regarding gender distribution of study participants, we would like to clarify that it is a general pattern in Pakistan that more 'male' are on administrative positions.

Comment 3.4: Study design - a more comprehensive description of the secondary healthcare systems would be beneficial to the reader. For example, what types of specialists? And are hospital admissions the same as a hospital? This lacks clarity for readers unfamiliar with the healthcare system.

Response 3.4: A brief description on Pakistani context has been added, including a more elaborated details on country's healthcare system. Please see page 5.

Comment 3.5: p.5 line 50 "secondary and TERRITORY-level" should this be "tertiary" instead?

Response 3.5: Thank you. This has been corrected.

Comment 3.6: Interviews - were the interviews conducted individually? please clarify why the provinces of Sindh and Punjab were targeted? - How was accuracy of translation into English tested?

Response 3.6: Yes, the interviews were conducted individually from all participants. A key reason for conducting this study in Sindh and Punjab was limited financial resources and they fact that these are the two most populous provinces in the country, and the impact of COVID19 – in terms of infected people – was relatively high as compared with the other two provinces (Baluchistan and Khyber Pakhtunkhwa). Regarding translation, we would like to highlight the tools were primarily translated in Urdu which is spoken and understood in Pakistan, especially among health care professional. To ensure the quality of translation, we made revision during the training of data collectors and even after initial pre-test exercise.

Comment 3.7: Quotes - consider placing in a table to reduce length
-unclear why some quotes are italicized and others are not
-consider inserting brackets [] in the quotes with words to aid the reader understanding (e.g. they would get hyper - is this upset or nervous?)

Response 3.7: Thank you. We have organised all the quotes and we now hope that they fit well into the narrative. Since there are single and multiple quotes within each theme, we are of the view that it will provide an easy and coherent story for the readers. If there are editorial requirements, we will be happy to shift these to a table. Secondly, all quotes are now italicised; this was an error. As per the suggestion of reviewer, square brackets have now been added to bring more clarity in the quotations.

Comment 3.8: It may be helpful to the reader to have a better understanding of he mental health of healthcare workers in Pakistan. What are the attitudes about mental health support among health professionals?

Response 3.8: Thank you. We have added a brief description on Pakistani context. As per your suggestion, this point has been added in that description. Please see page 17.

Comment 3.9: p. 6 line 16 should "Punjab providence" read "Punjab province"?

Response 3.9: Corrected.

Comment 3.10: Discussion - It may be helpful to emphasize the importance of workforce attitudes and expectations within Pakistan and how to deliver culturally acceptable workforce supports.

Response 3.10: Thank you. We have revised the discussion and more context specific points have been added as per your suggestion.

COI statements:

VERSION 2 – REVIEW

REVIEWER	Mostafa Bijani Fasa University of Medical Science, Department of Medical Surgical Nursing, Fasa University of Medical Sciences, Fasa, Iran
REVIEW RETURNED	16-Sep-2022

GENERAL COMMENTS	Dear editor, thank you for the opportunity to review the article again. The article is improved and worth publishing
---

REVIEWER	Michelle Zechner Rutgers University
REVIEW RETURNED	02-Oct-2022

GENERAL COMMENTS	The revised manuscript appears strengthened.
--